# Weaker plant-enemy interactions decrease tree seedling diversity with edge-effects in a fragmented tropical forest

Meghna Krishnadas [1], Robert Bagchi [2], Sachin Sridhara[3] & Liza S. Comita[1,4]

In fragmented forests, tree diversity declines near edges but the ecological processes underlying this loss of diversity remain poorly understood. Theory predicts that top-down regulation of seedling recruitment by insect herbivores and fungal pathogens contributes to maintaining tree diversity in forests, but it is unknown whether proximity to forest edges compromises these diversity-enhancing biotic interactions. Here we experimentally demonstrate that weakened activity of fungal pathogens and insect herbivores reduced seedling diversity, despite similar diversity of seed rain, during recruitment near forest edges in a human-modified tropical landscape. Only at sites farthest from forest edges (90–100 m) did the application of pesticides lower seedling diversity relative to control plots. Notably, lower seedling diversity corresponded with weaker density-dependent mortality attributable to insects and fungi during the seed-to-seedling transition. We provide mechanistic evidence that edge-effects can manifest as cryptic losses of crucial biotic interactions that maintain diversity.

[1] School of Forestry and Environmental Studies, Yale University, 195 Prospect Street, New Haven, CT 06511, USA. [2] Department of Ecology and Evolutionary Biology, University of Connecticut, 75N. Eagleville Road, Storrs, CT 06269, USA. [3] National Center for Biological Sciences, GKVK Campus, Bellary Road, Bangalore, Karnataka 560096, India. [4] Smithsonian Tropical Research Institute, Box, 0843-03092 Balboa, Ancón, Panama. Correspondence and requests for materials should be addressed to M.K. (email: meghna.krishnadas@yale.edu)

Discerning the ecological mechanisms that generate and maintain diversity of natural communities is fundamental to the informed management of ecosystems[1]. In plant communities, natural enemies (e.g., fungal pathogens and insect herbivores) have long been posited as mediators of a key mechanism that promotes diversity[2–4]. The Janzen–Connell hypothesis links natural enemies and plant diversity[3,5]—specialized pathogens and herbivores are more likely to damage or kill seeds and seedlings of a plant species, as that species becomes more abundant in the community. Such conspecific negative density dependence (henceforth CNDD) curtails recruitment of abundant plant species, prevents the exclusion of rare species, and thus promotes diversity[3,5,6]. Despite the well-established importance of insect and fungal enemies in structuring plant communities in a variety of ecosystems[4,7,8], how this mechanism for species coexistence is affected by human modification of natural habitats remains poorly known[9].

Human modification of forests can alter plant-enemy interactions in several ways. Change in environmental conditions closer to forest edges can modify fungal and insect activity[10,11]. For example, higher light availability near edges could decrease the efficacy of enemies by allowing recruiting seedlings to grow faster or recover from damage better than in the shaded forest understory[12–14]. Low moisture and humidity might also modify fungal and insect activity at forest edges compared with interiors[15], affecting patterns of seed and seedling predation. Moreover, habitat fragmentation negatively impacts host-specialized insects more than generalists[16,17]. Although a similar decline in host specificity of fungi with edge-effects has not been explicitly tested, edge-effects tend to favor habitat-generalist fungi[15,18]. Host-generalist enemies also mediate density-dependent recruitment but theory suggests that the effectiveness of enemies in promoting plant diversity—by advantaging species when rare[19]—increases with host specificity[20,21]. Notwithstanding proximate causes, if edge-effects weaken enemy-mediated density-dependent seedling recruitment, then species with more abundant seeds would prevail near edges, eroding diversity during seed-to-seedling transition[8].

Here we examine a central theoretical proposition, that natural enemies help shape plant diversity[4,6,8], in an applied context of urgent relevance—how does forest fragmentation modify the mechanisms that maintain plant diversity[6,9]? We expected that weaker top-down regulation of species recruitment by natural enemies would be associated with loss of seedling diversity near edges in fragmented, human-modified forests[6,9]. To test this hypothesis, we conducted in situ field experiments in a 3600 ha landscape matrix of tea plantations, roads, and fragments of wet tropical forest, located in the Western Ghats Biodiversity Hotspot, Karnataka, India (12°56'N and 75°39'E; Fig. 1a). As insects and fungi regulate community diversity most strongly during early life stages[4,8], we focused on plant-enemy interactions during the seed-to-seedling transition. We monitored seed arrival and seedling recruitment in adjacent plots within sampling stations arrayed at increasing distances from the forest edge. Along the edge-to-interior gradient, we then tested how insects and fungi regulated species recruitment and resulting diversity of recruits during the seed-to-seedling transition (Fig. 1b–d).

At 15 randomly chosen locations, we established 45 sampling replicates (hereafter stations) at three distances categories from the edge—0–5 m (E0), 20–30 m (E1), 50–60 m (E2)—and an additional 15 stations at 90–100 m from the edge (E3) where possible. Each station had two 1 $m^2$ seed traps made of 0.5 mm plastic mesh suspended 0.5–0.75 m above the ground and five 1 $m^2$ seedling plots. We randomly assigned seedling plots at each station to one of four treatments: control, fungicide, insecticide, and both fungicide and insecticide (see Methods). From

September 2015 until November 2016, we applied pesticide treatments every 10 days in the dry season (November–May) and every 5–7 days during the monsoons (June–October). We censused seed traps every 15 days and seedling recruits twice: at the end of the dry season (April 2016) and just after peak recruitment (November 2016).

We found that recruit diversity decreased from forest edge to interior, despite similar diversity of seed rain. In the interior-most sites, suppressing natural enemies with insecticide and fungicide reduced seedling diversity, and also corresponded with weaker density-dependent mortality during seed-to-seedling transition. In contrast, pesticide treatment up to 60 m of forest edges did not change seedling diversity or density-dependent mortality, suggesting a weaker role of this mechanism in structuring plant diversity near edges.

## Results

**Diversity and CNDD with distance-to-edge**. We first assessed whether diversity (inverse Simpson and exponentiated Shannon) of recruiting seedlings decreased closer to forest edges[22,23]. Lower diversity of recruits could stem from lower diversity of seed rain[24], but we found that seed diversity did not vary with distance to forest edge (inverse Simpson, Fig. 2a; exponentiated Shannon, Supplementary Figure 1a). Nonetheless, recruit diversity was significantly higher at sites 90–100 m from forest edges than at sites within 60 m of the edge (Fig. 2b, $Z_{141} = 2.61$, $P = 0.01$). Moreover, closer to edges, seedling assemblages were significantly less diverse than seeds that arrived, indicating a substantial erosion of diversity during the seed-to-seedling transition (Fig. 2a, b, Supplementary Figure 1b). Consistent with higher diversity of recruits in interior forest, the mean density-dependent mortality during the seed-to-seedling transition was strongest at sites 90–100 m from forest edge (Fig. 2c).

**Natural enemy effects on diversity and CNDD**. To determine whether the erosion of diversity closer to edges was due to altered plant-enemy interactions, we applied fungicide and insecticide treatments to a subset of plots at each sampling station (Fig. 1d). We observed an effect of pesticides on seedling diversity only at sites 90–100 m from edges (Fig. 3). Here, recruits were significantly less diverse, in terms of effective species number, in plots sprayed with fungicide ($Z_{141} = -3.51$, $P < 0.01$) compared with control plots. In contrast, within 60 m of forest edges, fungicide and insecticide treatments had little or no effect on seedling diversity (Fig. 3). Results were similar for the exponentiated Shannon's index (Supplementary Figure 2). Moreover, differences in diversity among treatments were not due to total seedling number, which was uncorrelated with either distance to edge or pesticide treatment (Supplementary Table 1 and Supplementary Table 2).

As with recruit diversity, only in the interior-most sites did density-dependent seed-to-seedling transition (CNDD) weaken by a statistically significant degree in fungicide and insecticide plots relative to control plots (Fig. 4, Supplementary Table 3a). However, contrary to expectation, mean community-wide CNDD was present, i.e., $b < 1$, at all distances from the edge.

## Discussion

For a given diversity of seeds arriving at a site, applying insecticide and fungicide treatments reduced the diversity of recruiting seedlings at 90–100 m from forest edge, but up to 60 m, seedling diversity in pesticide-treated plots did not differ from control plots. Moreover, only at 90–100 m did insecticide and fungicide diminish CNDD during seed-to-seedling transition. Notably, we found that pesticides altered recruitment rates at all distances

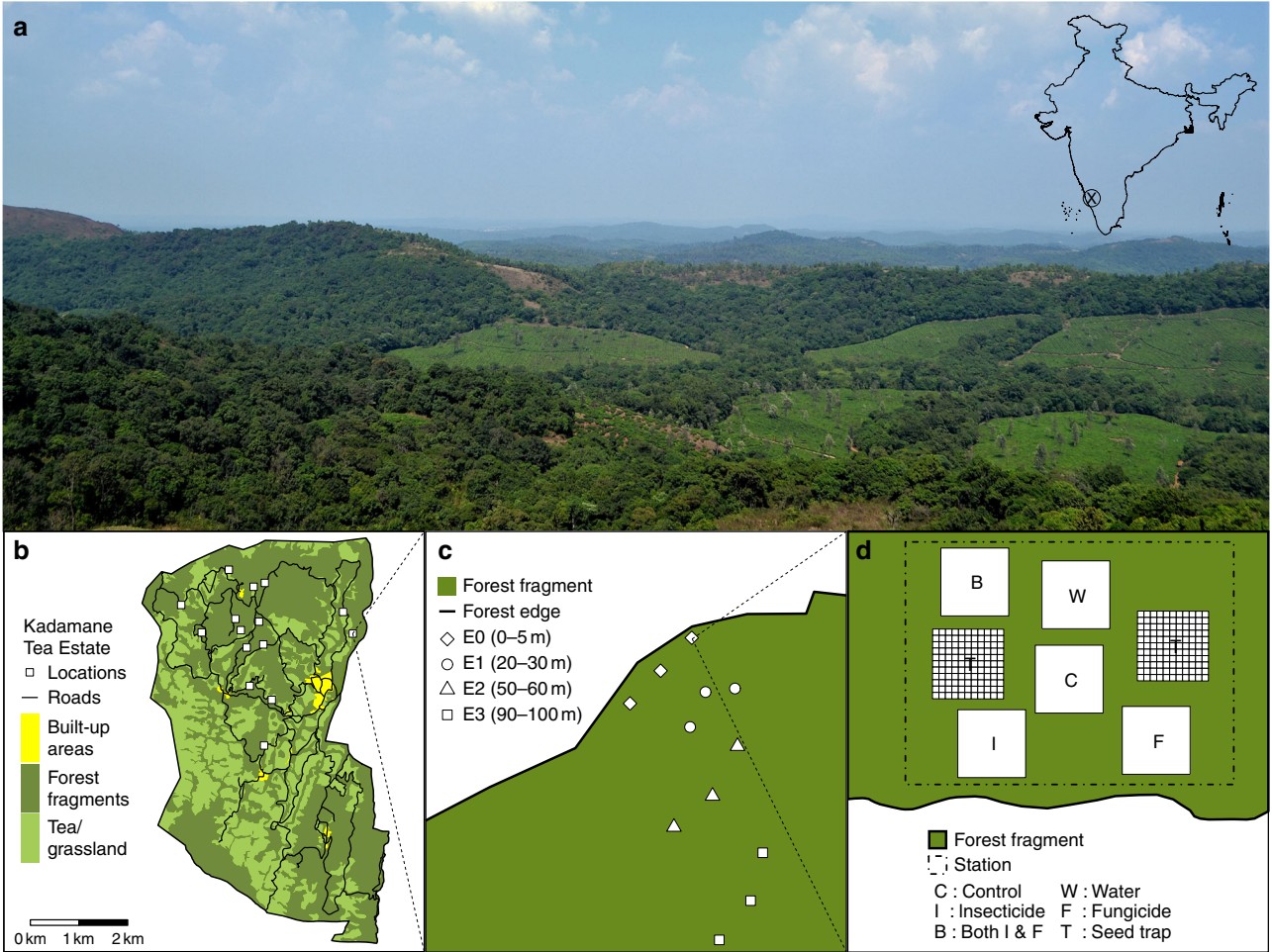

**Fig. 1** Study landscape and sampling design. **a** The research site in the Western Ghats Biodiversity Hotspot in southwest India (inset): a matrix of forest fragments interspersed with tea plantations and roads. **b** Map of study area and sampled locations. **c** Experimental design to test how edge-effects modified plant–enemy interactions and associated diversity of seedlings. Groups of sampling stations were placed at the following distance categories from the forest edge: E0 = 0–5 m, E1 = 20–30 m, E2 = 50–60 m, and E3 = 90–100 m. **d** Each station was comprised of two seed traps and five seedling plots: two control plots (C, W) and one each treated with either insecticide (I), fungicide (F), or both pesticides (B)

from the edge (Supplementary Table 4), indicating that proximity to edges did not obviate natural enemy activity. Together, these results indicate that proximity to edges reduced the contribution of natural enemies to promoting diversity during seedling recruitment. It is puzzling though that the combined effect of both pesticides did not reduce diversity and CNDD more than each pesticide alone, suggesting the need to investigate possible interactions between insect and fungal activity for plant diversity. Suppressing both insects and fungi could also increase the role of other microbial pathogens (e.g., bacteria) in mediating diversity, or enhance the influence of mycorrhizal fungi on seedling recruitment[4].

Notably, CNDD occurred at all distances from the forest edge (Fig. 2c). Seed mortality due to generalist natural enemies (e.g., some rodent granivores, vertebrate browsers, or other micro-organisms) could increase near forest edges[25,26], which could explain the CNDD we found at edges and why this CNDD was not altered by fungicide and insecticide (Supplementary Table 3a). However, as the diversifying effect of natural enemies would be more likely via host-specific predators[20], our results suggest the need to examine whether edge-effects alter the nature of plant–enemy interactions, e.g., by reducing specialized interactions relative to generalized ones[17]. For example, there are barely any studies that show whether effective specialization of

fungal pathogens changes with edge-effects or if fungal communities on different hosts become more homogeneous near edges. Specialized plant–fungal interactions are notoriously hard to quantify for the whole plant community, but molecular techniques can help assess whether specialization between hosts and enemies changes near forest edges[17,27].

Sites at 90–100 m (E3) also experienced the largest compositional change due to shifts in species abundances during the seed-to-seedling transition (Supplementary Figure 3). The seed and seedling community were most dissimilar at E3 and only fungicide significantly reduced this dissimilarity. These differences suggest that in addition to increasing niche differences, fungi promote seedling diversity by also changing the relative abundances of species during the seed-to-seedling transition, i.e., by modifying competitive ability[28]. In comparison, insects affected seedling recruitment relatively more independent of species abundances, perhaps explaining their smaller effect on diversity than fungal pathogens[8]. Compared with fungi, insect activity may be restricted by their own parasites or predators[29], which may explain why these two enemies differed in their contribution to CNDD and compositional shifts.

Shifts in species' relative abundances could be related to species natural history[14]. Smaller-seeded species tend to increase in the seed rain at forest edges compared with interiors[24], but higher

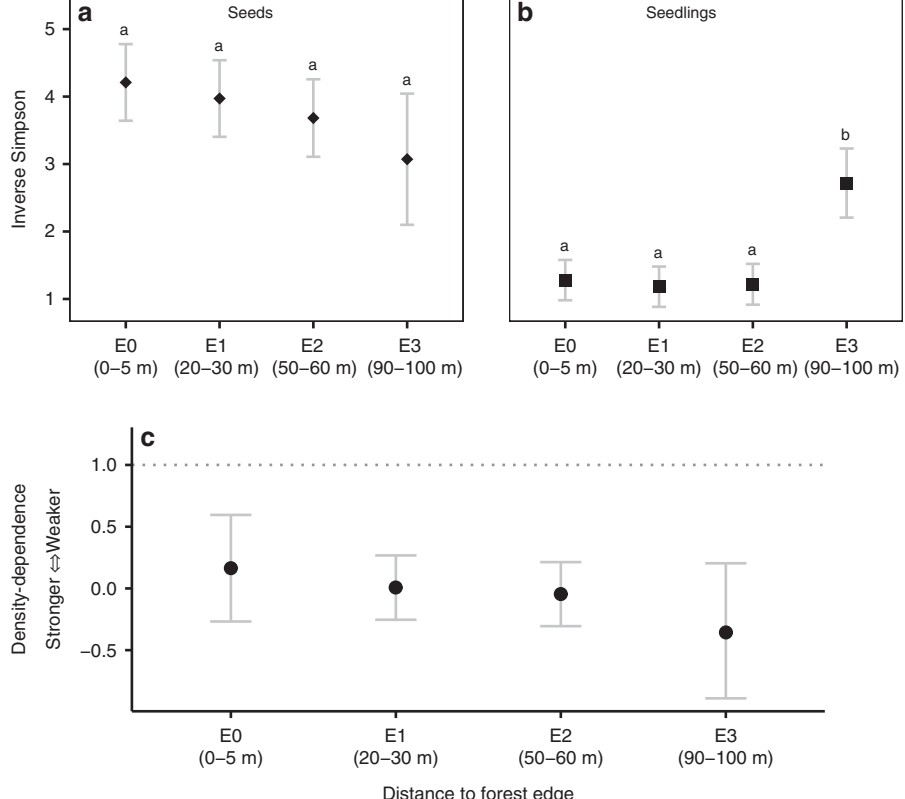

**Fig. 2** Variation in seed and seedling diversity and CNDD at different distances from the forest edge. Inverse Simpson diversity of **a** seeds arriving in seed traps ($N = 146$) and **b** seedlings that recruited into adjacent 1 m² control plots ($N = 146$) at increasing distances from forest edge. Larger values indicate higher diversity. Points represent mean values per edge-distance and error bars show 95% confidence intervals for the means. E0 through E3 represent increasing distances from the forest edge. Letters denote significant pairwise differences in seed and seedling diversity among edge categories estimated using linear mixed-effects models. Panel **c** depicts mean strength of conspecific density dependence during seed-to-seedling transition observed in control plots ($N = 146$) at increasing distances from the forest edge. Points represent strength of CNDD estimated using hierarchical Bayesian analysis and error bars provide 95% Bayesian credible intervals. Dotted line represents no CNDD (slope = 1)

seed production in smaller-seeded species often correlates with poorer defenses against natural enemies[30]. Weaker enemy effects at edges could allow species with abundant seeds to maintain their high numbers compared with interior forest where natural enemies would decrease recruitment rates of abundant seeds and seedlings[14]. Moreover, common species tend to have greater diversity of pathogen resistance genes than rare species[31]. Generalized natural enemy effects might benefit common species more than rare species overall. Alternatively, species that are rare in intact forests because of depredation by natural enemies might increase with weaker enemy activity, resulting in loss of genetic diversity against disease.

Change in seedling diversity in forest fragments could result from abiotic changes near forest edges[23]—either directly or via weakened impacts of natural enemies. Elevated light levels are a particularly prominent edge-effect in fragmented tropical forests. Seedling recruitment rates can vary with light availability[30] or light can indirectly affect seedling diversity by altering the strength of CNDD[12,14,32]. In this forest however, median light availability did not vary with distance to edge (Supplementary Figure 4). Moreover, heterogeneity in light availability across the landscape did not explain changes in seedling diversity or CNDD, whether by itself or in interaction with edge-distance and pesticide treatment (Supplementary Table 3b and Supplementary Table 5).

Gradients in moisture and pathogens could interactively regulate seedling recruitment in fragmented forests[33], although one

recent study found that composition of plant hosts structures fungal communities more than changes in soil moisture[27]. Moisture gradients can directly influence seedling species' distributions[34], but it is unclear to what degree small-scale differences in soil moisture affect diversity[35]. We did not directly measure soil moisture, but canopy openness can represent differences in local soil moisture and humidity, and we found no correlation between canopy openness and seedling diversity (Supplementary Table 6). However, at our study site, most recruitment occurs during the monsoons when moisture is unlikely to be a limiting factor for germination and recruitment. Nevertheless, in seasonally wet tropical forests, edge-effects could alter seedling composition by intensifying the effects of dry season water stress on seedling establishment and survival[36].

Disrupting the feedbacks between plants and their natural enemies is theorized to decrease plant diversity[4,21], but this mechanism has not been demonstrated in human-modified forests. Consistent with theoretical expectations[3,6,9], we found that weaker plant-enemy interactions during seed-to-seedling transition corresponded with reduced diversity of recruiting seedlings. Such loss of diversity during plant recruitment can compound the declines in diversity from habitat loss[37] and elevated mortality of trees in fragments[38], with implications for future composition and function of fragmented forests. Considering that nearly 20% of the world's remaining forests are within 100 m of an edge[39], our results suggest that cryptic dilution of mechanisms that enable species coexistence might be widespread and could pose a

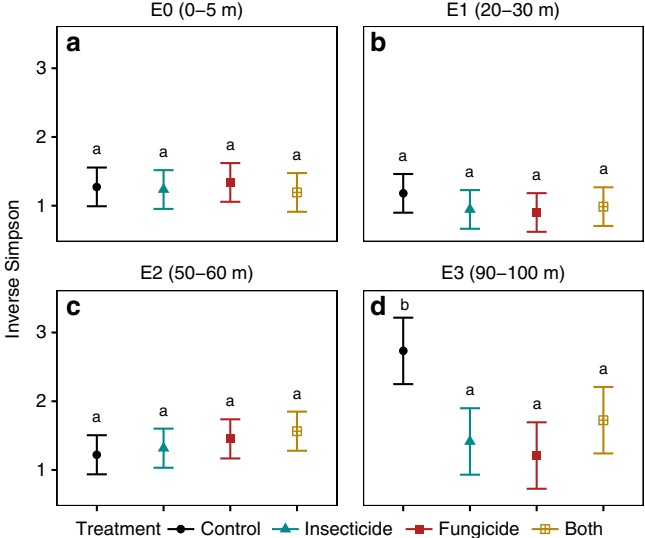

**Fig. 3** Change in seedling diversity with pesticide treatments at increasing distances from the forest edge. Diversity was estimated for: **a** 0–5 m (E0), **b** 20–30 m (E1), **c** 50–60 m (E2), and **d** E3 (90–100 m) from the forest edge. Points represent mean observed diversity of new seedlings recruiting into 1 m$^2$ plots where the different pesticide treatments were applied ($N = 146$ per treatment). Error bars show 95% confidence intervals. Statistical differences were estimated using linear mixed-effects models, letters denote significant pairwise differences among edge-treatment combinations within each distance category

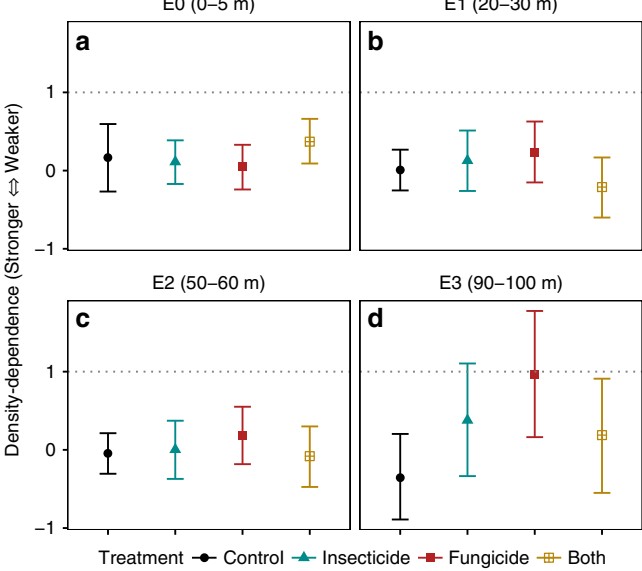

**Fig. 4** Change in mean strength of CNDD with pesticide treatment at increasing distances from the forest edge. CNDD was estimated at: **a** 0–5 m (E0), **b** 20–30 m (E1), **c** 50–60 m (E2), and **d** E3 (90–100 m) from the forest edge. Points represent estimated strength of CNDD ($N = 146$ per treatment) and bars provide 95% Bayesian credible intervals for parameters estimated using generalized linear mixed-effects models in a hierarchical Bayesian framework. Dotted line represents no CNDD (slope of density-dependent seed-to-seedling transition = 1). The degree of overlap in credible intervals is an indication of the strength of difference between treatments—lesser overlap suggesting stronger differences. Separate slopes were estimated per species and group-level intercepts per species and station

long-term threat to sustaining biodiversity in fragmented forests. Discerning the mechanistic underpinnings of ecological processes that maintain species diversity can help manage and restore biodiversity and ecosystem function in human-modified forests[23].

## Methods

**Study site**. We conducted this study within Kadamane Tea Estate (henceforth Kadamane), a fragmented forest landscape in the Western Ghats biodiversity hotspot, Karnataka, south India (12°56'N and 75°39'E). The study area is a ca. 3600 ha matrix of tea, abandoned coffee, roads, grassland, and fragments of wet tropical forest. Clear felling to establish tea fields occurred from 1920s through 1930s and no large-scale forest conversion occurred thereafter. Some parts of the remaining forests were selectively logged for timber species (mainly *Mesua ferrea*, *Hopea canarensis*, *Elaeocarpus tuberculatus*, *Dimocarpus longan*, and *Actinodaphne bourdilloni*), until banned by the Supreme Court in 1996. Sporadic logging continued until 2002. Today, biomass removal is limited to collecting small stems for fuelwood and construction. Forest fragments constitute about 60% of the study area (ca. 2300 ha). Forest edges have recovered to a closed canopy and many edges have a regenerating wall of vegetation. Forests are primarily on shallow slopes and valleys in this undulating terrain, with grasslands on ridge tops and steep slopes (Fig. 1). Mean annual rainfall in the region is ca. 5000 mm, most of which occurs from June through September. Soils are predominantly well-drained, clayey Alfisols derived from a gneiss base.

**Sampling scheme**. We first demarcated all forest fragments within Kadamane using existing topographic maps. We used open source Geographical Information Systems tool Quantum GIS (QGIS, version 2.17) to pick 45 random locations at the edges of forest fragments using the following criteria: (1) locations were at the edge of forest patches that were separated from an adjoining patch by at least 20 m, (2) locations were in forest patches with at least 100 m to the center, and (3) locations were at least 250 m apart. We then physically surveyed these 45 locations and excluded sites that were in topographically distinct areas such as hilltops, associated with markedly different habitat conditions such as large streams, inaccessible, heavily logged or otherwise altered, or where distance from edge was associated with marked habitat transitions such as large streams, rocky substrate, human-made clearings, tree falls or canopy gaps. Of the remaining 23 locations, we selected the 15 least disturbed in terms of signs of human activity (without any trails or cut stems). The straight-line distance between closest pairs of locations was 350–560 m (mean = 468 m). If it was not possible to lay sampling stations at the chosen point, we moved 50 m to the left or right along a line parallel to the edge.

Existing evidence suggests that compositional changes in seedling communities occurs largely within 50 m of forest edges[23]. Hence, we established sampling stations at three distance categories from the edge—0–5 m (E0), 20–30 m (E1), and 50–60 m (E2). We laid three replicates per site per edge-distance, totaling 45 replicates per distance category. To capture the interior-most conditions, we established 15 replicates at 90–100 m from the edge (E3), ensuring that we fulfilled all other criteria for site selection (elevation, substrate, human-use intensity). We were logistically limited to 15 replicates for E3, because distances of 90 + meters in all other locations were situated in topographically steep or rocky areas where the tree community was markedly different in structure and composition, and we wanted to control for variation in elevation and topography. There was also no large contiguous forest nearby. This situation reflects the reality of typical human-modified landscapes in the tropics, where remnant forests are confined to steep and inaccessible areas. However, patterns of diversity and CNDD in relation to edge-distance × pesticide interactions were qualitatively similar when we restricted our analysis to the subset of sites that had interior-most locations (5 sites × 3 replicates per site = 15 replicates for each distance category), supporting the validity of our results (diversity: Supplementary Table 7, CNDD: Supplementary Figure 5).

**Seed and seedling data**. At each distance category, we placed three circular sampling stations parallel to the edge with their centers 20–25 m apart. Within each station, we established five 1 × 1 m seedling plots, demarcated at two diagonal corners using PVC (plastic) pipes, and placed two seed traps made of 1 m$^2$ pieces of 0.5 mm PVC mesh, suspended 0.5–1 m above the ground and secured to nearby stems. Each location had 9–12 stations with 15 seedling plots per distance category, totaling 135–190 plots (Fig. 1). In total, we laid 146 stations and 730 seedling plots across the landscape, after broadly controlling for elevation, topography, and human disturbance.

**Natural enemy exclusion experiments**. To evaluate the role of fungal and insect pathogens in mediating species recruitment during seed-to-seedling transition, we randomly assigned three of the five seedling plots at every station to be sprayed with one of three treatments: fungicide, insecticide, or both fungicide and insecticide. For the fungicide treatment, we used a combination of Amistar® (Syngenta Ltd, active ingredient: azoxystrobin) and Ridomil Gold® (Syngenta Ltd; active ingredients: mancozeb and metalaxyl). Amistar® offers broad-spectrum systemic protection against multiple plant pathogenic fungi, and Ridomil® acts against oomycetes and fungi. Both fungicides have low toxicity to non-target organisms

and minimally inhibited arbuscular mycorrhizal fungi in temperate grasslands, crops, and in seedlings of two Neotropical tree species (ref. [40] and references therein). We used Actara® (Syngenta Ltd, active ingredient: thiamethoxam) as an insecticide treatment, which provides systemic and contact protection against a broad range of insects. We applied pesticides with a hand mister following manufacturer's guidelines (Amistar: 0.01 g, Ridomil: 0.5 g, Actara: 0.021 g, each dissolved in 100 ml water per 1 $m^2$ plot). One control plot was sprayed with 100 ml of water and the last plot was left untreated (control). During preliminary analysis, we found no significant differences in diversity and seedling number between control and water treatments so for all subsequent analysis we used their mean value as Control. From September 2015 until November 2016, we applied pesticide treatments every 10 days during the dry season and every 5–7 days during the wet season. To avoid any run-off from agriculture, we set up sampling stations upslope of tea fields. Most forest remnants were located upslope, because these fragments are preserved to maintain watersheds for the estate. Also, to prevent pollution of the river systems that provide water for drinking and domestic use, this tea estate does not engage in large-scale application of pesticides, only using copper sulfate on shoot tips and at the base of plants.

**Seed rain and seedling census.** We collected data on seed rain and seedling recruitment from September 2015 until November 2016. We recorded seed fall into seed traps at stations every 10–15 days and recorded only viable seeds without any obvious mechanical or pathogen damage. We classified seeds into morphospecies and prepared a photo-catalog. Species were then identified from literature, online seed databases, consultation with local botanists, and field observations of fallen fruits and seeds underneath adult trees. Before we began pesticide treatments of seedling plots, we conducted an initial seedling census where all existing seedlings were tagged with colored plastic rings and given unique IDs based on combinations of plot ID, color, and tag number. Thereafter, we conducted one census at the end of the dry season (March 2016) and a final census after the recruitment peak that follows the rainy season (November 2016), in which we tagged all new seedlings. Seedlings were identified based on field observations and help from local botanists. Over the course of the experiment, we matched seedlings to corresponding seeds using field observations of seeds germinating under parent trees and by growing seedlings from seeds in the greenhouse. We were able to match 91% of observed seedling species to their seeds.

**Canopy openness and light levels.** We characterized canopy openness (correlated with moisture and humidity) and understory light availability using hemispherical photos taken 0.25 m above ground at the center of each 1 $m^2$ seedling plot with a digital camera (Nikon Coolpix 950, Melville, NY, USA) fitted with a fish-eye lens (Nikon FC-E8, Melville, NY, USA). We took photographs during early morning from mid-June through mid-July, which is the beginning of the rainy season and skies are uniformly overcast. Images were analyzed using Gap Light Analyzer[41]. We conducted preliminary analyses with multiple metrics separately. Here we provide results from models using percent canopy openness (% Canopy. Open) and total transmitted radiation (% Trans. Tot.), because they provided the best fit for models of diversity and CNDD.

**Statistical analysis.** We used linear mixed-effects models with Gaussian errors to analyze diversity of new seedling recruits in relation to an interaction between distance to forest edge and pesticide treatment, and included seed diversity as an additive effect. To account for variability in baseline diversity due to location, we modeled group-level intercepts per station. We used the model estimates to test two hypotheses: (1) diversity of seedling recruits increased farther from forest edges; (2) closer to edges, seedling diversity did not change with pesticide application but pesticides decreased seedling diversity in forest interiors. We used a similar modeling approach to test how diversity varied with light × pesticide interaction. To ensure that results were not dependent on the diversity metric used, we ran all analyses with three different, commonly used diversity indices: rarefied species richness (with two individuals sampled for seedlings and five for seeds), inverse Simpson, and exponentiated Shannon.

We estimated CNDD using species found as seeds or seedlings in at least ten stations across the landscape and had at least threefold variation in seed density. We also excluded species with fewer than five seedlings across all sampled sites. For the 22 remaining species, we modeled community-wide strength of CNDD using the power-law relationship describing seedling number in relation to seed number per species[8,42]. The relationship between number of seeds (**S**) and seedling recruits (**R**) at a station is described by $R = a * S^b$, where $a$ is the baseline probability of seed-to-seedling transition and $b$ is the strength of density dependence (CNDD)[42]. Thus, $b = 1$ implies no CNDD; smaller $b$ values represent stronger CNDD. We used the estimates of $b$ per edge-distance × pesticide treatment combination to test two hypotheses: (1) CNDD was weaker closer to forest edges and (2) CNDD weakened in plots sprayed with pesticides, but only farther from forest edges.

To model strength of CNDD during seed-to-seedling transition, we used generalized linear mixed-effects models with negative binomial errors, with random intercepts and slopes per species to account for interspecific variation[43,44]. Identity of sampling stations was also included as a random intercept to account for spatial differences in location. As seed rain and seedling recruitment were

measured in adjacent plots, but not in the same plot, the potential for spatial mismatch can introduce a measurement error that biases the slope of seed–seedling relationship toward zero and leads to spuriously strong estimates of CNDD[6]. We corrected for measurement error by modeling the number of seeds in a trap as drawn from a log-normal distribution parameterized by the mean and variance of seeds observed in two adjacent seed traps. We modeled this multi-level analysis for seed-to-seedling transition in a hierarchical Bayesian framework, which allows incorporation of natural variability and data uncertainty when predictor variables are inaccurately measured. In addition, posterior distributions for parameters capture the potential variation in effect sizes of predictor variables[45]. In the few cases where seedlings of a species were recorded at a station but not its seeds, we set mean seed number and seed variance = observed seedling number. This introduces a conservative bias against detecting CNDD[42]. We used a similar modeling framework to test how CNDD varied with an interaction between light and pesticide treatment.

We used program R version 3.3.2[46] for all analysis. We used package vegan[47] to calculate diversity indices and package nlme[48] to model diversity in relation to edge × pesticide interactions. For Bayesian analysis, we used package brms[49] in program R to interface with Stan version 2.17.0[50].

## Data availability
The datasets generated and analyzed during the current study and corresponding computer code are available from the corresponding author on reasonable request.

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

## Acknowledgements

The Garden Club of America, Harvard Arnold Arboretum, Yale Tropical Resources Institute, and Yale Institute of Biospheric Studies funded this research. We thank Kadamane Estate Company for permitting use of their property for research and Mr Cariappa for logistical support. We thank Raghunath from Nature Conservation Foundation for sharing Kadamane map layers. Meghana R, Kavya Agarwal, and Arun Kumar helped with data collection. M.K. thanks Ajith Kumar for his support. This study was made possible by the dedication of our field assistants—Netraprasad Sharma, Suresh Roy, and Kanjimalai.

## Author contributions

M.K. developed the project, collected the data, performed analysis, and wrote the first draft of the manuscript with inputs from L.S.C. and R.B. at all stages. S.S. aided data in collection and analysis. All authors discussed the results and provided comments.

## Additional information

**Competing interests:** The authors declare no competing interests.

