## [Peer Review File · Nature Communications]

Reviewers' comments:

Reviewer #1 (Remarks to the Author):

Overall, this paper is a solid contribution to our understanding of how edge effects alter the negative feedbacks that maintain plant diversity. The authors set up an extensive field experiment in a fragmented landscape in which they measured seed rain and seedling recruitment at different distances from the forest edge, then treated a subsample of plots with fungicide, insecticide, both, or neither. They found that only in their most interior plots (90-100 m from the edge) did biocides decrease seedling diversity and weaken density dependence. This is a very interesting result highlighting how anthropogenic changes to forest habitats can alter diversity-maintaining interactions between plants and their enemies. I have some general comments about the interpretation of the data and the conclusions drawn from these results.

One thing that confused me as I read this manuscript was the use of the term "diversity." The authors repeatedly discussed changes in diversity while displaying figures that showed species richness (Figure 2a and b, Figure 3) or discussing results about rarefied species richness. The authors have tested the same hypotheses using rarefied species richness as well as two different diversity indices, and those results are displayed as Supplementary Figures S1 and S2. Though all of these results are qualitatively similar, not all are exactly the same – for example, compare Figure 3d with S2d and S2h (which the authors discuss lines 102-104); or the results of Table S7a with S7b. There also might be something to these differences in diversity vs. richness – maybe due to some key changes in abundance? It's worth exploring. Regardless, I think they should clearly differentiate between richness and diversity throughout the body of the manuscript.

The authors placed some importance on differences between effects on "generalized" and "specialized" natural enemies, but this argument needs some clarification. Do they mean "generalized" enemies in terms of host affinity? Environmental tolerance? All of the above? The traditional Janzen-Connell models emphasize host specificity. But one of the three references cited (Mueller et al 2016) discusses fungal specialization in terms of environmental niche breadth rather than host identity (and in a very broad swath of fungi, not just pathogens). So it's not clear to me where the authors are coming from here. Further, I am not sure what evidence that they have points towards a change in the nature of the natural enemies from edge to interior in terms of a decrease in specialists (Lines 120-122). Yes, there is an increase in CNDD in the interior, but who these enemies are is not really quantified or discussed much beyond very broad categories. So I think the authors need to take more care to consider this potential mechanism in light of the data they have as well as the information they lack.

I also think the authors could give a little more consideration to the role of soil moisture in mediating NDD. For example, wet conditions can be more conducive to fungal activity, so while water may not be limiting to plants in this site (Lines 144-146), excess water can lead to more disease. True, canopy openness can be a proxy for soil moisture but that doesn't really replace on-the-ground measurements. I don't think the authors need to have the mechanism for their results figured out – but I do think the manuscript would be improved by allowing for a couple of different explanations to be at play. After all, abiotic and biotic influences on NDD are not mutually exclusive.

Conversely, an interesting piece of data that is not emphasized enough is the fact that CNDD is prevalent in both edges and interiors, which speaks to the overall importance of CNDD in diversity maintenance even in light of human modification. I'm curious if there are differences in the strength of CNDD regardless of treatment between the four different edge habitats -- are the differences between the four points in Figure 2C statistically significant? It doesn't appear so from eyeballing the figure, but I'm curious if that is formally borne out in the analyses. Either way, that is also an interesting result.

A couple of other questions to follow up on:

Figure S5: It looks like, when a subset of plots is analyzed, density dependence also disappears with the fungicide/insecticide treatment right on the edge (Figure S5a). How would you explain this?

Table S3: Why do you think that insecticides and fungicides, when applied separately, had overall positive effects on recruitment but not the combination of the two?

Figure 3d, S3d: Along the same lines, any thoughts on why the differences between fungicide and other biocide applications are different here? You broach this in lines 115-117, but don't really offer a reason why that incorporates differences between these two groups of organisms.

Though the manuscript itself needs some modifications, the authors present some quite compelling data. I'd just like to see it explored a little more. The paper is well written and edited, so I only have a few minor suggestions listed below:

Line 17: Contributes not contribute

Line 40: It's not sure how you would bridge a proposition – consider rephrasing

Line 87: Which component(s) of Figure 2?

Reviewer #2 (Remarks to the Author):

The manuscript by Krishnadas et al. presents a very nice and novel study that investigates the effects of pest and pathogens on seedling recruitment along tropical rain forest edges. Several ecological processes and mechanisms governing seedling recruitment are affected within forest fragments and along their edges, as is the nature of biotic interactions. The present experiment is well designed, the analysis appropriate and the results are sound, and of great interest for tropical rainforest biologists, ecologists and conservationists. The study provides a clear and sound methodology that could be replicated in other tropical rainforests around the world. There are, however, few aspects in my opinion, that need to be incorporated into the discussion. Seed and seedling damage by pests and pathogens is a common interaction in the wet tropics, however, there are some traits of seedling species that makes them less and/or more susceptible to herbivore and fungal attack. Therefore, I suggest including the information (if available) or to in a few lines discuss the following aspects: (a) for instance, what species were more likely to be affected by pests and pathogens, common or rare seedling species? And (b) Host traits (e.g., late vs. early successional plant species) have been shown to differentially affect plant susceptibility to pest and pathogens. In other words, biotic damage could be driven by the life history traits of seedlings (Coley 1980; Coley & Barone 1996). Considering that early successional seedling species are more likely to recruit along rain forest edges, how this condition could have affected your findings?

Other suggestions (L=line number in the manuscript):

L95-97. Should be moved to the Methods Section

L138. Please clarify, how plant hosts shape fungal pathogen communities?

Reviewers' comments:

Reviewer #1 (Remarks to the Author):

Overall, this paper is a solid contribution to our understanding of how edge effects alter the negative feedbacks that maintain plant diversity. The authors set up an extensive field experiment in a fragmented landscape in which they measured seed rain and seedling recruitment at different distances from the forest edge, then treated a subsample of plots with fungicide, insecticide, both, or neither. They found that only in their most interior plots (90-100 m from the edge) did biocides decrease seedling diversity and weaken density dependence. This is a very interesting result highlighting how anthropogenic changes to forest habitats can alter diversity-maintaining interactions between plants and their enemies. I have some general comments about the interpretation of the data and the conclusions drawn from these results.

Thank you for your constructive and helpful feedback.

One thing that confused me as I read this manuscript was the use of the term “diversity.” The authors repeatedly discussed changes in diversity while displaying figures that showed species richness (Figure 2a and b, Figure 3) or discussing results about rarefied species richness. The authors have tested the same hypotheses using rarefied species richness as well as two different diversity indices, and those results are displayed as Supplementary Figures S1 and S2. Though all of these results are qualitatively similar, not all are exactly the same – for example, compare Figure 3d with S2d and S2h (which the authors discuss lines 102-104); or the results of Table S7a with S7b. There also might be something to these differences in diversity vs. richness – maybe due to some key changes in abundance? It’s worth exploring. Regardless, I think they should clearly differentiate between richness and diversity throughout the body of the manuscript.

We apologize for any confusion in our use of the word ‘richness’. We only examined ‘rarefied species richness’, which is a measure of diversity that accounts for differences in species abundances and is not equivalent to raw species richness (i.e. number of species observed), which is sensitive to sample size. Our intention in this manuscript was to focus on diversity, because CNDD is expected to act by reducing the abundance of common species in the community. Reduced abundance of common species would, in turn, allow a larger number of species to coexist by preventing rare species from being excluded from the community. However, at the temporal and spatial scales examined in our study (i.e. seed-to- seedling transition at the 1-m² scale), we expected to see changes in species abundances due to CNDD, but not necessarily changes in species number.

To make it clear that we are looking at diversity, we have removed the results for ‘rarefied species richness’. We think that the slight difference in results with rarefied richness vs. inverse Simpson and Shannon might be due to the fact that we rarefied for two individuals (minimum number of individuals found in the seedling plots). This essentially translates into Simpson’s

diversity, i.e. the probability that any two individuals belong to different species. Because this probability can be sensitive to species' abundances and rare species, especially in relatively less speciose communities such as seedlings within 1-m² plots, we have decided to only present results for inverse Simpson and exponentiated Shannon. Both these indices take into account species richness and abundances but are not as sensitive to sample size as rarefied richness. We present the inverse Simpson as our main result because it translates into effective species number and is less sensitive to rare species than exponentiated Shannon's index.

The authors placed some importance on differences between effects on “generalized” and “specialized” natural enemies, but this argument needs some clarification. Do they mean “generalized” enemies in terms of host affinity? Environmental tolerance? All of the above? The traditional Janzen-Connell models emphasize host specificity. But one of the three references cited (Mueller et al 2016) discusses fungal specialization in terms of environmental niche breadth rather than host identity (and in a very broad swath of fungi, not just pathogens). So it's not clear to me where the authors are coming from here. Further, I am not sure what evidence that they have points towards a change in the nature of the natural enemies from edge to interior in terms of a decrease in specialists (Lines 120-122). Yes, there is an increase in CNDD in the interior, but who these enemies are is not really quantified or discussed much beyond very broad categories. So, I think the authors need to take more care to consider this potential mechanism in light of the data they have as well as the information they lack.

We emphasized the role of host-specific enemies because theoretical work indicates that the diversifying effects of natural enemies increases with their host-specificity. However, as the reviewer correctly noted, we do not actually test whether edge-effects change insect and fungal abundances with respect to their host-specificity. Such an experiment was beyond the scope of this study, but this would be an interesting follow-up to our results. We have now toned down the argument about generalized vs. specialized interactions. Line 114 – 120:

“Notably, CNDD occurred at all distances from the forest edge (Fig 2c). Seed mortality due to generalist natural enemies (e.g. rodent granivores, vertebrate browsers and non-fungal microorganisms) could increase near forest edges^{22,23}, which could explain the CNDD we found at edges and the fact that this CNDD was not altered by applying fungicide and insecticide. However, since the diversifying effect of natural enemies would be more likely via host-specific predators¹⁶, our results suggest the need to examine whether edge-effects alter the nature of plant-enemy interactions, e.g. by reducing . specialized interactions relative to generalized ones¹⁴.”

I also think the authors could give a little more consideration to the role of soil moisture in mediating NDD. For example, wet conditions can be more conducive to fungal activity, so while water may not be limiting to plants in this site (Lines 144-146), excess water can lead to more disease. True, canopy openness can be a proxy for soil moisture but that doesn't really replace on-the-ground measurements. I don't think the authors need to have the mechanism for their results figured out – but I do think the manuscript would be improved by allowing for a couple of different explanations to be at play. After all, abiotic and biotic influences on NDD are not mutually exclusive.

We agree with the reviewer's point that biotic and abiotic drivers of recruitment are not mutually exclusive. We have now discussed in more detail the potential role of soil moisture for spatial variation in recruitment and in interacting with fungal disease. Lines 152 – 167.

Conversely, an interesting piece of data that is not emphasized enough is the fact that CNDD is prevalent in both edges and interiors, which speaks to the overall importance of CNDD in diversity maintenance even in light of human modification. I'm curious if there are differences in the strength of CNDD regardless of treatment between the four different edge habitats -- are the differences between the four points in Figure 2C statistically significant? It doesn't appear so from eyeballing the figure, but I'm curious if that is formally borne out in the analyses. Either way, that is also an interesting result.

As the reviewer correctly notes, CNDD did not vary by a statistically significant degree in control plots from edge to interior, even though CNDD appeared to become stronger towards the interior. We think this could have occurred because of rodent seed predation (personal observation). Another possibility is the role of other microbial enemies. We have now discussed these points in Lines 114 – 120.

A couple of other questions to follow up on:

Figure S5: It looks like, when a subset of plots is analyzed, density dependence also disappears with the fungicide/insecticide treatment right on the edge (Figure S5a). How would you explain this?

Yes, this is an interesting occurrence, particularly with regard to the treatment with both biocides. These sites were in the fragments that had the largest amount of ecologically similar habitat. There could be a fragment-size dependent effect on natural enemy activity, i.e. the largest fragments tend to maintain more natural enemy activity even at edges. Two of these five sites were located at edges separated by canopy breaks created by a 25-30 m wide road and not tea, and it is possible that edges next to tea fields suffered a greater decline in natural enemy activity. Overall, this would have led to a smaller/non-significant change in CNDD in control vs. treatments when all locations were analyzed together. Edges were also more variable in their microclimate than interiors (Fig S4), which might explain some of the differences in the results for a subset of the locations.

Table S3: Why do you think that insecticides and fungicides, when applied separately, had overall positive effects on recruitment but not the combination of the two?

We are not sure why applying both pesticides did not compound the effects seen with each pesticide separately. It is possible that curtailing both fungi and insects increased the activity of other natural enemies, e.g. rodents and bacteria. However, we did not discuss this result in detail because we did not find enough information to come up with a reasonable argument and wanted to avoid being overly speculative.

Figure 3d, S3d: Along the same lines, any thoughts on why the differences between fungicide and other biocide applications are different here? You broach this in lines 115-117, but don't really offer a reason why that incorporates differences between these two groups of organisms.

The difference between fungicide vs. insecticide and both pesticides can be driven by how these two enemies regulate species abundances during recruitment. Previous studies have found that the action of insects may be less density-dependent than fungi. We have now discussed this point in lines 127 - 130:

“The difference between fungicide and insecticide treatments suggest that fungi promote diversity by changing the relative abundances of species during the seed-to-seedling transition. In comparison, insects affected seedling recruitment relatively more independent of species abundances, perhaps explaining their smaller effect on diversity than fungal pathogens⁸.”

Though the manuscript itself needs some modifications, the authors present some quite compelling data. I'd just like to see it explored a little more.

Thank you for the constructive comments and insightful feedback. We have tried to address all your points and hope that you will find the revised manuscript satisfactory.

The paper is well written and edited, so I only have a few minor suggestions listed below:

Line 17: Contributes not contribute

Changed.

Line 40: It's not sure how you would bridge a proposition – consider rephrasing

Rephrased to: “Here, we examine a key theoretical proposition, that natural enemies help shape plant diversity^{4,6,8}, in an applied context of urgent relevance—how does forest fragmentation modify the mechanisms that maintain plant diversity^{6,9}?”

Line 87: Which component(s) of Figure 2?

Now mentioned.

Reviewer #2 (Remarks to the Author):

The manuscript by Krishnadas et al. presents a very nice and novel study that investigates the effects of pest and pathogens on seedling recruitment along tropical rain forest edges. Several ecological processes and mechanisms governing seedling recruitment are affected within forest fragments and along their edges, as is the nature of biotic interactions. The present experiment is well designed, the analysis appropriate and the results are sound, and of great interest for tropical rainforest biologists, ecologists and conservationists. The study provides a clear and sound methodology that could be replicated in other tropical rainforests around the world.

Thank you for the positive feedback and helpful comments. Please see our responses below.

There are, however, few aspects in my opinion, that need to be incorporated into the discussion. Seed and seedling damage by pests and pathogens is a common interaction in the wet tropics, however, there are some traits of seedling species that makes them less and/or more susceptible to herbivore and fungal attack. Therefore, I suggest including the information (if available) or to in a few lines discuss the following aspects:

(a) for instance, what species were more likely to be affected by pests and pathogens, common or rare seedling species?

This is a good question, since common and rare species can differ in their sensitivity to CNDD and natural enemies. One potential cause of the differences between rare vs. common species might be that lower sensitivity to pathogens that mediate CNDD make species more “common”. We have now discussed life-history based differences in plant response to natural enemies. Lines 131 – 141.

However, we would like to highlight that this data set does not allow for analysis of individual species, especially rare species, making it difficult to directly test for differences among common vs. rare species without many more years of data or more extensive sampling to increase sample sizes for individual species.

(b) Host traits (e.g., late vs. early successional plant species) have been shown to differentially affect plant susceptibility to pest and pathogens. In other words, biotic damage could be driven by the life history traits of seedlings (Coley 1980; Coley & Barone 1996). Considering that early successional seedling species are more likely to recruit along rain forest edges, how this condition could have affected your findings?

The differential response of host species to natural enemies is definitely an important feature that drives CNDD dynamics. In an experimental test of edge x pathogen interaction for four species, two early successional vs. two long-lived pioneers, we found that the early successional species experienced weaker CNDD at the forest edge (Krishnadas and Comita, 2017). Additionally, in a separate analysis that we are submitting, we have examined how species traits that correspond with differences in species' life-histories explain interspecific differences in recruitment rates

with pesticide application. We think that including and discussing the analysis on interspecific differences would detract from the main message of this paper, which is about diversity. However, we have now discussed the implications of how life-history based differences in plant response to natural enemies could affect diversity at forest edges. Lines 131 – 136.

“Shifts in species’ relative abundances could be related to species natural history²⁵. Smaller-seeded species tend to increase in the seed rain at forest edges compared to interiors²⁰, but higher seed production in smaller-seeded species often correlates with poorer defenses against natural enemies²⁶. Weaker plant-enemy interactions at edges could allow species with abundant seeds to maintain their high numbers compared to interior forest where natural enemies would decrease recruitment rates of abundant seeds and seedlings²⁵. Moreover, common species might be common because they have a greater diversity of pathogen resistance genes than rare species²⁷. Weaker natural enemy effects might benefit common species more than rare species overall. Alternatively, species that are ‘rare’ in intact forests because of depredation by natural enemies might increase with weaker enemy activity, resulting in loss of genetic diversity against disease.”

Other suggestions (L=line number in the manuscript):
L95-97. Should be moved to the Methods Section

We agree that this line seems more appropriate for the Methods section. However, given the Nature Communications format of having detailed Methods only at the end of the paper, we thought that including this information here would help orient the reader towards how we tested our hypothesis regarding plant-enemy interactions and recruit diversity. We have therefore retained this information in its current position.

L138. Please clarify, how plant hosts shape fungal pathogen communities?

We based our argument on a recent paper by Sarmiento et al. (2018, PNAS) where they found that fungal community composition was correlated with the composition of host plants on which different fungal species appeared to specialize, but fungal composition did not vary with change in soil properties. Of course, this result is from a single site and may not be true everywhere. Now changed to:

“Gradients in moisture and pathogens could interactively regulate seedling recruitment in fragmented forests³⁰, although one recent study found that composition of plant hosts structure fungal communities more than changes in soil moisture²⁴.”

REVIEWERS' COMMENTS:

Reviewer #1 (Remarks to the Author):

For the most part, I think the authors addressed my main concerns in their revision and in the corresponding letter. I appreciate their thoughtful responses, and only have a few brief comments:

Lines 44-46: You may want to add some references to papers that explore how light affects CNDD – there are quite a few out there.

Line 46: "Environmental differences...": Such as?

Lines 48-49: Reference 13 is a fungal paper, but the text is only about changes in insect communities.

Line 51: Similarly, the text is about patterns in fungal communities but the reference is about insects.

Line 113: AMF are typically considered mutualists, not pathogens (though their overall impacts can fall on a spectrum from mutualistic to parasitic).

Lines 114-117: "Seed mortality due to generalist natural enemies (e.g. rodent granivores, vertebrate browsers and non-fungal microorganisms) could increase near forest edges..." This language is a little confusing and needs to be clarified. It makes it sound as if all non-fungal microbes are generalists, which is not necessarily the case.

Line 153: This reference again does not match the text.

Figure S2: Legend needs to be changed to reflect that only the results using the Exponentiated Shannon index are now shown here.

Reviewer #2 (Remarks to the Author):

Edge effects can influence several aspects of tropical rain forests regeneration, but little is known about how edge effects affect seedling recruitment and the interaction between seeds, seedlings and their natural enemies, namely insects pests and fungal pathogens. Insect herbivores and fungi help to maintain plant species diversity but also influence plant's ability to survive, grow, reproduce and compete. The present study showed that both types of natural enemies are affected by human disturbances and that their incidence differently affected the seed to seedling transition across fragment's edges and interiors. To my knowledge, this consequence of edge creation has been yet unexplored. Therefore, as previously stated, Krishnadas et al. presented a very nice and novel well designed study. In my opinion the authors have properly addressed all the reviewers comments and corrections. I have just one observation, what about the friendly fungi, mycorrhizas? Or "friendly" insects that help seed germination by breaking the seed coat? Authors have stated that both fungicides used in the study are known to minimally inhibit arbuscular mycorrhizal fungi, however, this fact deserves further investigation (not right now!!) and discussion. In other words, the experimental application of fungicides and insecticides is likely to alter some aspects of these complex biotic interactions and therefore, seed germination and seedling recruitment.

RESPONSE TO REVIEWERS:

Reviewer #1 (Remarks to the Author):

For the most part, I think the authors addressed my main concerns in their revision and in the corresponding letter. I appreciate their thoughtful responses, and only have a few brief comments:

Lines 44-46: You may want to add some references to papers that explore how light affects CNDD – there are quite a few out there.

Added.

Line 46: “Environmental differences...”: Such as?

There seems to have been an oversight in Lines 42-46 during the last revision. We had already mentioned environmental differences in general in line 43 and followed up with specific examples in subsequent lines. However, we mistakenly retained the phrase ‘environmental differences’ in line 46. Now changed to:

“Low moisture and humidity might also modify fungal and insect activity at forest edges compared to interiors¹², affecting patterns of seed and seedling predation.”

Lines 48-49: Reference 13 is a fungal paper, but the text is only about changes in insect communities.

Reference 13 is Rossetti et al. Ecology Letters, and examines only changes in insect communities with fragmentation. Just to be sure, we have double-checked all the citations in this section.

Line 51: Similarly, the text is about patterns in fungal communities but the reference is about insects.

The reviewer appears to have misunderstood our goal of the paragraph and we apologize if we were not clear. We wanted to emphasize that the ability of natural enemies to promote diversity would decrease with reduced host-specificity. To illustrate this point, we have used general theoretical papers as well as references that deal with specific groups of natural enemies. From the literature, it appears that the effects of fragmentation have been examined more for insects than fungi, and we highlight this gap. We emphasize the importance of host-specific pathogens using conceptual papers, e.g. Benitez et al. 2013, TREE.

Indeed, fungal host-specificity is a future topic of investigation suggested by our results. There are barely any studies which show whether effective specialization of fungal pathogens changes with edge-effects or if fungal communities on different hosts become more homogeneous near edges.

Line 113: AMF are typically considered mutualists, not pathogens (though their overall impacts can fall on a spectrum from mutualistic to parasitic).

Changed.

Lines 114-117: “Seed mortality due to generalist natural enemies (e.g. rodent granivores, vertebrate browsers and non-fungal microorganisms) could increase near forest edges...” This language is a little confusing and needs to be clarified. It makes it sound as if all non-fungal microbes are generalists, which is not necessarily the case.

We agree. We have now changed this to read:

“Seed mortality due to generalist natural enemies (e.g. some rodent granivores, vertebrate browsers or generalized microorganisms) could increase near forest edges^{23,24},”

Line 153: This reference again does not match the text.

Corrected.

Figure S2: Legend needs to be changed to reflect that only the results using the Exponentiated Shannon index are now shown here.

Changed.

Reviewer #2 (Remarks to the Author):

Edge effects can influence several aspects of tropical rain forests regeneration, but little is known about how edge effects affect seedling recruitment and the interaction between seeds, seedlings and their natural enemies, namely insect pests and fungal pathogens. Insect herbivores and fungi help to maintain plant species diversity but also influence plant’s ability to survive, grow, reproduce and compete. The present study showed that both types of natural enemies are affected by human disturbances and that their incidence differently affected the seed to seedling transition across fragment’s edges and interiors. To my knowledge, this consequence of edge creation has been yet unexplored. Therefore, as previously stated, Krishnadas et al. presented a very nice and novel well designed study. In my opinion the authors have properly addressed all the reviewers comments and corrections.

Thank you for your constructive feedback.

I have just one observation, what about the friendly fungi, mycorrhizas? Or “friendly” insects that help seed germination by breaking the seed coat? Authors have stated that both fungicides used in the study are known to minimally inhibit arbuscular mycorrhizal fungi, however, this fact deserves further investigation (not right now!!) and discussion. In other words, the experimental application of fungicides and insecticides is likely to alter some aspects of these complex biotic interactions and therefore, seed germination and seedling recruitment.

The reviewer makes a good point about mutualistic and helpful fungi and insects. Indeed, plant recruitment and diversity likely arise from the relative impacts of positive and negative effects between plants and their consumers. We have briefly alluded to this in lines 112-117. However, we have not elaborately discussed this because, admittedly, these multi-trophic interactions can

be complex and it can be hard to pin point the exact changes due to applying pesticides. Furthermore, we were interested in mean, community-level effects of suppressing natural enemies and our results suggest that overall, weaker effects of natural enemies might be contributing to the lower diversity found at edges.